# How to Defend COVID-19 in Taiwan? Talk about People’s Disease Awareness, Attitudes, Behaviors and the Impact of Physical and Mental Health

**DOI:** 10.3390/ijerph17134694

**Published:** 2020-06-30

**Authors:** Chin-Hsien Hsu, Hsiao-Hsien Lin, Chun-Chih Wang, Shangwun Jhang

**Affiliations:** 1Department of Recreation and Sports Management, National Chin-Yi University of Technology, Taichung 41170, Taiwan; hsu6292000@yahoo.com.tw; 2Department of Recreation and Holistic Wellness, Mingdao University, No. 369, Wen-Hua Rd., Peetow, Changhua 52345, Taiwan; rg500ww@yahoo.com.tw; 3Division of Neurosurgery, Department of Surgery, Changhua Christian Hospital, Changhua 50006, Taiwan; 133393@cch.org.tw

**Keywords:** Coronavirus Disease 2019, prevention measures, Taiwan

## Abstract

This study explored awareness, attitudes, and behavior in relation to Coronavirus Disease 2019 (COVID-19) prevention among Taiwanese citizens and their physical and mental health statuses. Through collection of 2132 questionnaire responses in field research, the present researchers analyzed the data using descriptive statistics and various approaches. In conclusion, the public’s high level of willingness to share information, sufficient knowledge of and consensus on epidemic prevention between individuals and families, strict compliance with relevant regulations, effective preventive measures, and adequate public facilities have contributed to control of COVID-19. However, vigilance and awareness of the pandemic in some individuals, epidemic-prevention campaigns, and community-based preventive measures were insufficient. Some citizens subsequently suffered from headaches, anxiety, and mood instability. Furthermore, demographic variables (place of residence, sex, age, and occupation) and physical and mental health status produced various effects on citizens’ awareness, attitude, and behavior regarding epidemic prevention as well as the perceived effect of COVID-19 on physical and mental health.

## 1. Introduction

Coronavirus Disease 2019 (COVID-19) was first discovered in Wuhan, China, although the origins of the virus remain unknown. After the first case was confirmed on 26 December 2019 in Wuhan, the virus rapidly spread, causing a global pandemic. On 9 April 2020, the accumulated number of confirmed cases in China reached 83,249 with a death toll of 3344 in 105 days [1,2]. The daily number of confirmed cases and deaths averaged 826 and 31.8, respectively, indicating the severity of the COVID-19 pandemic. Presently, the pandemic has affected 183 countries with 1,457,989 confirmed cases and 86,730 deaths worldwide [3]. With a mortality rate of 5.95% [4], the virus can damage pulmonary alveoli, leading to respiratory failure and death in severe cases. Currently, the origin of COVID-19 is unknown. COVID-19 can live in the air for a long time, and infected individuals with and without symptoms can both spread the disease [5]. In addition, the transmission of this virus remains unclear. COVID19, the global economy has been substantially affected [6,7]; the pandemic’s severity and the damage subsequently caused are difficult to estimate. Figure 1 illustrates the regions affected by COVID-19 worldwide and in Taiwan.

Despite its proximity to China, Taiwan has successfully controlled COVID-19 because of its geographical advantages and experience gained from the epidemic of severe acute respiratory syndrome. Since the disease outbreak on 21 January 2020, the Taiwanese government has actively adopted various measures to contain the virus spread [8], instructed its citizens in how to protect themselves, controlled the healthcare supply chain, and encouraged academic institutions to develop antiviral drugs. By 9 April, 79 days after the first case was identified, the accumulated numbers of confirmed cases and deaths were 379 and 5, respectively [9], averaging 4.7 cases and 0.06 deaths per day. These figures are substantially lower than those in China, where the number of confirmed cases is 175.74 times the number in Taiwan and the number of deaths 5300 times the number in Taiwan. The numbers of confirmed cases and deaths in Taiwan rank 70th and 92nd, respectively among the 183 affected countries (including developed, developing, and underdeveloped countries) [10]. These results indicate that the measures against COVID-19 adopted by Taiwan are effective and can thus provide a reference for other countries [11]. Figure 2 demonstrates the numbers of confirmed cases in Taiwan.

In addition to the government, citizens play a crucial role in pandemic control. Citizens must comply with the policies implemented by the government, demonstrated awareness through psychological conceptualization, perception, judgement, and imagination. They are then able to develop appropriate attitudes toward control measures, develop appropriate awareness, and take appropriate actions [12,13,14,15]. Under such circumstances, the country has the ability to withstand tangible and intangible pressures resulting from the pandemic, respond to global challenges, and adopt appropriate epidemic prevention measures. To assess awareness of epidemic prevention, citizens’ knowledge—including individual, family, and community health awareness and compliance with government instructions [13,16]—can be observed, along with the country’s relevant measures and emergency responses, including notifications on protective measures [15]. Attitudes toward the disease can be determined through observation of individuals’ perceptions and explicit reactions [17,18,19], such as their vigilance and research in relation to epidemic prevention and health [15]. One can determine individuals’ behaviors in relation to the disease by evaluating their adaptability and personal conduct during the pandemic [20], [21]. For example, research can examine citizens’ vigilance regarding epidemic prevention, their hygiene, their problem-solving skills, and public information [15].

Citizens must maintain their health to achieve epidemic prevention [22]. When individuals experience favorable physical, mental, and social health [23] and their mental states enable them to recognize their own potentials, meet the needs of normal life, and contribute to society [24], their immune systems are likely to improve [25], thereby reducing stress and, subsequently, risks of physical or mental illnesses [26,27]. Therefore, the physical–mental health of citizens is an essential factor in epidemic prevention. To understand an individual’s physical and mental health, researchers should explore his or her physiological, psychological, and spiritual status [28,29], such as emotions and feelings in everyday life or in the workplace, job performance, physical health and reflections, and general attitude toward life.

COVID-19 has infected numerous people worldwide. Each country strives to adopt effective measures to tackle the pandemic; prevent transmission; reduce infection risks; restore the economy, protect society, and preserve environments; and offset losses caused during the pandemic [30]. However, the vaccine for this disease has not yet been developed. Therefore, learning from countries that have successfully contained the virus can help governments to make appropriate decisions and overcome current challenges [31]. This study explored Taiwanese citizens in terms of their awareness, attitude, and behavior in relation to COVID-19 prevention, and their physical–mental health. The present researchers then analyzed influential factors influencing the effectiveness of measures imposed by the government. The results provide a reference for the Taiwanese government and governments worldwide to implement and adjust measures against the pandemic.

## 2. Methods and Instruments

### 2.1. Study Framework and Hypotheses

This research aimed primarily to study awareness, attitudes, and behaviors in relation to COVID-19 prevention among the Taiwanese public as well as their physical and mental health. Subsequently, the researchers compared citizens living in different regions by comparing the numbers of confirmed cases in these areas. Finally, the study analyzed how the government and people in Taiwan have successfully controlled the infection. Taiwan has 22 cities and counties, which are usually divided into five areas according to their locations, populations, resources, and economic activities [32], as shown in Table 1.

In these five regions, we investigated awareness, attitude, and behavior in relation to epidemic prevention among citizens, as well as their physical and mental health. The results helped the research team to understand the public’s awareness of epidemic prevention, government measures for raising such awareness, and approached to prevention of the disease’s transmission in Taiwan. The research questionnaire was designed on the basis of other studies [3,4,5,6,7,8,9,10,11,12,13,14,15,16,17,18,19,20,21,22,23,24,25,26,27,28,29,30,31,32,33,34,35,36,37,38]. The researchers collected data by conducting field research, individual interviews and questionnaire surveys [39]. Through data matching and assessment [40], summarization, and analysis, the researchers prepared this manuscript [41]. With the collected data, this study investigated the effects of citizen awareness, attitudes, behaviors, and physical and mental health on epidemic prevention. Figure 3 depicts the framework of this study.

**Hypothesis** **1 (H1).**
*Citizens’ awareness, attitudes, and behaviors in relation to epidemic prevention are inconsistent with their physical and mental health.*


**Hypothesis** **2 (H2).**
*Citizens’ awareness attitudes, and behaviors in relation to epidemic prevention among citizens living in various regions are inconsistent with their physical and mental health statuses.*


### 2.2. Study Procedure and Instruments

Facing the overwhelming global pandemic, the government and the people need to take appropriate actions [12,13,14,15] in which the public needs to have a correct understanding of the virus and fully cooperate with the epidemic prevention measures to effectively defend against the threat of disease. Additionally, it also requires citizens to exercise to keep themselves in good shape, physically and mentally. It was found that most of the current COVD-19 studies explore the symptoms caused by the virus, the impects on economic development, the public health risks, and other policy-related issues [3,4,5,6,7,8]. Not many studies have been conducted on the public’s perceptions of the epidemic and defensive behaviors from their perspective. Studies on people’s self-assessment of their physical and mental health are also few and those that adopt the spatial concepts for analysis are even rarer.

This study originates in the realization that there is a lack of prior studies that explore the epidemic prevention awareness and the physical and mental health issues of the public during the COVD-19 epidemic by applying the concept of spatial distribution. Furthermore, although there have been relevant reports focusing on the self assessment of health cognition, attitudes, behaviors, as well as physical and mental health, none of these studies were conducted in the context of the global pandemic. Researchers will refer to relevant literature [3,4,5,6,7,8,9,10,11,12,13,14,15,16,17,18,19,20,21,22,23,24,25,26,27,28,29,30,31,32,33,34,35,36,37] was reviewed, and the results were examined on the basis of grounded theory [42,43,44,45]. The questionnaire items taken from [12,13,14,15] and [35,36,37] were revised to devise 14 items on epidemic prevention awareness, 8 on epidemic prevention attitudes, and 16 on epidemic prevention behaviors. Following [32] and [35,36,37], 16 items on physical and mental health were included.

Five scholars or experts with diverse backgrounds in public health, medicine, recreational sports, and healthcare were invited to help in refining the questionnaire. Respondents answered each item with a score ranging from 1 to 5 according to their attitudes or beliefs about the dimension or topic, with 1 representing minimal feeling and 5 the strongest feeling. In February 2020, 50 copies of the questionnaire were distributed for a pretest, and the results were analyzed using SPSS for Windows 22.0. When KMO > 0.08 and the p-value in Bartlett’s test is less than 0.01 (*p* < 0.01), this indicates that the scale is suitable for continuing to conduct factor analysis [46], and then the coefficient alpha is greater than 0.80 after the test, indicating that this questionnaire has good reliability [47]. 

According to the statistical analysis results, there are 14 questions about epidemic prevention awareness. The value of KMO is 0.967, χ2 value yielded from the Bartlett test is 33919.717, df is 91, and the significance value is 0.000 (*p* < 0.001), suitable for conducting factor analysis. The explained variances of the scale are 70.78%, 13.631%, and 2.783%, and the total explained variance is 77.193%. After factor analysis, all of them are retained, and they are named Epidemic prevention awareness (4), Consensus on decision-making (4), and Emergency response (6) respectively, 3 dimensions containing a total of 14 questions. The α coefficients of the three scales are 0.970, 0.966, and 0.968 respectively, and the α coefficient of the total scale is 0.970. According to the above analysis results, we can know that this questionnaire has good reliability.

There are 8 questions about epidemic prevention attitudes. The value of KMO is 0.938, χ2 value yielded from the Bartlett test is 21,256.586, df is 28, and the significance value is 0.000 (*p* < 0.001), suitable for conducting factor analysis. The explained variances of the scales are 78.696% and 5.071%, and the total explained variance is 83.767%. After factor analysis, all of them are retained, and they are named Perception (4) and Explicit reaction (4), 2 dimensions containing a total of 8 questions. The α coefficients of the two scales are 0.961 and 0.966 respectively, and the α coefficient of the total scale is 0.966. According to the above analysis results, we can know that this questionnaire has good reliability.

There are 16 questions about epidemic prevention behaviors. The value of KMO is 0.967, χ2 value yielded from the Bartlett test is 38,088.237, df is 120, and the significance value is 0.000 (*p* < 0.001), suitable for conducting factor analysis. The explained variances of the scales are 67.489% and 6.201%, and the total explained variance is 73.69%. After factor analysis, all of them are retained, and they are named Daily life adaptability (11), Individual performance (4), 2 dimensions containing a total of 8 questions. The α coefficients of the two scales are 0.967 and 0.969 respectively, and the α coefficient of the total scale is 0.970. According to the above analysis results, we can know that this questionnaire has good reliability.

There are 16 questions about influences on physical and mental health. The value of KMO is 0.953, χ2 value yielded from the Bartlett test is 30574.351, df is 120, and the significance value is 0.000 (*p* < 0.001), suitable for conducting factor analysis. The explained variances of the scales are 55.796%, 11.104%, and 2.697%, and the total explained variance is 69.597%. After factor analysis, all of them are retained, and they are named Psychological status (5), Spiritual status (5), and Attitude and health (6), 3 dimensions containing 16 questions. The α coefficients of the three scales are 0.948, 0.943, and 0.946 respectively, and the α coefficient of the total scale is 0.948. According to the above analysis results, we can know that this questionnaire has good reliability.

In summary, based on the above analyses, we can know that all dimensions of epidemic prevention awareness, epidemic prevention attitudes, epidemic prevention behavior, and influences on physical and mental health meet the test requirements, indicating that this questionnaire can be continued to be used for this study, as shown in Table 2.

Since the discovery of this phenomenon in January 2020, the researchers began to review relevant literature and develop a research process and framework. The samples were collected between March 1 and April 10, 2020. At the beginning of the sampling period, as the epidemic was unfolding, it was difficult to collect questionnaire information personally due to the government order and health concerns. Therefore, the DoSurvey online platform was used to retrieve questionnaire information. The participants were required to provide identification information while submitting their responses. As the epidemic eased down gradually from April 1, researchers conducted supplementary surveys using the field survey method. In addition to visiting each site to understand the development of the epidemic in each region, it also allowed the use of the sampling method to simultaneously collect information from the locals. A total of 2132 questionnaires were obtained and the results were compiled using the SPSS for Windows 22.0 statistical package.

In order to obtain more in-depth information, the researcher collected and analyzed questionnaire samples, and then designed the semi-structured interview content for the interview survey. After asking for people’s consent and consent, using Line and Faecbook social networking software interviewed a total of 10 scholars, owners, general public and students from different regions with health and hygiene expertise or research experience to provide them with sample analysis Insights into the results.

After the participants verified the accuracy of the recorded content, the researchers integrated the information of the questionnaire, analyzed the results, and cmpleted the research paper through the processes of induction, organization and analysis [41]. Finally, a multivariate validation analysis method was adopted to combine the information obtained from different research subjects, theories, and methods, to validate multiple data from multiple perspectives, and compare the results of different studies [39,40,42,43,44] in order to acquire accurate knowledge and implications, and to investigate the current status of public awareness, attitudes, and behaviors associated with epidemic prevention, and their perceptions of physical and mental health, as shown in Table 3.

### 2.3. Study Scope and Limitations

The study investigated the impact on epidemic cognition, attitudes, behaviors, as well as physical and mental health during the period of COVIS-19 by targeting the local residents in different regions of Taiwan. However, in the midst of epidemic prevention, out of the concern of the risk of asymaptomatic transmission of the disease [6,7], the government advocated policies and guidelines such as avoiding going out, participating in crowd activities, and taking public transportation, as well as maintaining one meter distance between people, and taking body temperature and wearing masks at all times [8]. The public’s fear of infection further reduced their willingness to contact with people [48] and decreased opportunities for interpersonal interaction thereby greatly affecting the sample collection.

Therefore, in order to successfully collect enough samples, a survey using online platform was performed. However, due to the above-mentioned restrictions and people’s concerns, as well as the constraints in transportation and funding, the difficulty in coordinating work and interview times, and the different experiences in using the online survey platform, people’s willingness to be interviewed were affected, which further influenced the information and proportion of the sample presentation. Although the associated limitations may impact the findings, we believe that there is still research value after supplementing the information with the multivariate research approach [39,40,42,43,44]. Hence, if the above discrepancies occur, we will provide corrections and suggestions for subsequent studies.

## 3. Analysis of Results

### 3.1. Demographics

People in Central Taiwan are friendly, and most of them exercise habitually. Because of a change in ideas regarding procreation, the area has seen a sharp rise in its female population [46]. Although the students in this area have sufficient free time, their intentions to travel are low due to budget restraints. Most of the respondents were residents of Central Taiwan (52.9%), female (60.5%), aged ≤ 20 years (50.1%), and were college or university students (79.6%). Most exercised and routinely engaged in recreational activities but had few opportunities to travel (69.4%). Most of the respondents reported satisfactory physical and mental health (91.9%).

Since the first confirmed diagnosis, the Taiwanese government has paid considerable attention to the potential threat of the COVID-19 epidemic. In addition to establishing the Central Epidemic Command Center, the government monitors the epidemic status in the entire country and applies responsive measures accordingly. It utilizes television, the Internet, and social media to broadcast news on the current epidemic status and epidemic prevention information for people to reference in preventing the disease. Many of the participants had acquired information on epidemic prevention from the Internet (44.8%) and television (46.4%), as shown in Table 4.

### 3.2. Analysis of Awareness, Attitudes, and Behaviors in Relation to Epidemic Prevention among Taiwanese Nationals and of Their Physical and Mental Health Statuses

Compared with other cases [35,36,37], the government actively invests in health and hygiene education, attaches importance to the people’s body and mind, and helps strengthen immunity in Taiwan [24,25,26]. Tis was a reason for the success of prevention in Taiwan, setting an example for other countries [11]. Descriptive statistics were applied to determine Taiwanese citizens’ physical and mental health as well as their awareness, attitudes, and behaviors in relation to epidemic prevention. The results revealed influential factors leading to the success of prevention measures adopted by the Taiwanese government, as listed in Figure 4.

Effects on citizens’ awareness, attitude, and behavior regarding pandemic prevention and their physical and mental health varied among regions, supporting Hypothesis 1. We discovered that consensus between individuals and families, public facilities, epidemic-prevention knowledge, and individual preventive actions were adequate. People were willing to share information with one another and follow government instructions. However, epidemic-prevention awareness, epidemic-prevention campaigns, vigilance in relation to the pandemic, and community hygiene were insufficient. Some individuals were subsequently affected by physical and mental health problems such as headaches, anxiety, and mood instability, as shown in Table 5.

Compared with their counterparts, people living in Northern Taiwan demonstrated greater concerns regarding the cleanliness of surroundings. The northern region is the major transportation hub, where most imported cases were confirmed. Moreover, prevention information was slightly insufficient, and some individuals had poor knowledge and vigilance of epidemic prevention. The accumulated number of confirmed cases in Taiwan was 379. Despite a few domestic cases, Taiwan has controlled the disease under control because of favorable individual and family prevention consensus, comprehensive prevention facilities in public spaces, strict compliance with individual and public prevention policies, mask wearing among most citizens, and willingness to share information [49]. Although the interviewees felt uncomfortable due to mental stress, Taiwan did not restrict people from going out [50], so their daily lives and work performance were not affected. Therefore, the epidemic control in Taiwan is stable.

### 3.3. Analysis of Awareness, Attitudes, and Behaviors in Relation to COVID-19 Prevention among Citizens in Various Regions and of their Physical–Mental Health

On the basis of the country’s experience with the 2003 SARS outbreak, the Taiwanese government established an independent health administration, and each county or city government set up an official health management branch for medical management and healthcare knowledge promotion, thereby reinforcing national health education and fostering a satisfactory sense of leisure and healthcare practices. In addition, the government manages and controls surgical mask distribution through convenience stores and pharmacies across Taiwan, and it utilizes social media platforms to transmit epidemic information in real time. However, due to the gap between urban and rural development in Taiwan, the level of health and hygiene education and the implementation of decision-making are different [50], and regional resources and lifestyles are different, which affects the effectiveness of epidemic prevention.

The success of epidemic prevention depends on efforts by individuals, families, and communities and their compliance with imposed measures and instructions [15]. Consensus between individuals and communities helps to improve policy effectiveness [22,23,24,25,26,27,28,29,30,31]. The effectiveness of prevention measures varied among cities and counties; some cities even had zero confirmed cases. However, patterns regarding the numbers of confirmed cases did not vary according to economic, social, and environmental characteristics in different regions. To descriptive statistics, the *t* test, and analysis of variance (ANOVA) were performed to compare the effects of place of residence, sex, age, occupation, and physical–mental health status on participants’ awareness, attitude, and behavior regarding COVID-19 prevention and the perceived effects on their physical and mental health.

#### 3.3.1. Northern Area

As presented in Table 6, for epidemic prevention awareness, the highest score was in family habits and consensus (4.3) and the lowest in epidemic-prevention campaign and the effectiveness of such advocacy in maintaining health (3.9). With attitudes, cooperation with policies had the highest score (4.2) and epidemic prevention information gathering the lowest (3.9). For behaviors, voluntarily wearing masks received the highest score (4.2) and community cleaning the lowest (3.6). Regarding physical and mental health, they were the most notable for anxiety and headaches (3.2) but least notable for suicide ideation (2.0). These results are partially consistent with findings in [35,36,37].

Further analysis revealed no significant differences according to occupation. However, women reported significantly (*p* < 0.01) stronger feelings than men in the following items: for attitudes, epidemic-prevention campaign, family habits, self-epidemic prevention, health protection, and public measures; for awareness, media information, teachers’ advice, public measures, and epidemic prevention information searching; regarding behaviors, epidemic prevention steps, cooperation, seeking solutions, wearing masks, and health management. Men reported stronger feelings (*p* < 0.01) for the following items: on physical and mental health, anxiety, headache, eating disorder, and feeling lost.

The respondents significantly differed by age (*p* < 0.01) in their scores on the following items: for awareness, campus epidemic prevention and teachers’ advice; on behaviors, seeking help from teachers, families, or friends, sharing information, and sense of justice. Those aged ≤50, ≥51, and 51–60 years were particularly sensitive to the items related to attitudes, awareness, and behaviors, respectively.

The reported physical and mental health of respondents significantly (*p* < 0.01) affected their scores for the following items: for attitudes, public consensus, health protection, and public measures; regarding awareness, campus epidemic prevention, media information, and public measures; on behaviors, epidemic prevention steps, cooperation, seeking solutions, voluntarily wearing masks, sharing information, and sense of justice. Compared with other respondents, those infected with COVID-19 exhibited low scores in all the awareness, attitude, and behavior dimensions and reported more health problems such as insomnia, indigestion, and overeating; the public also reported higher pain sensitivity, insomnia, sense of loss, and suicidal tendency. 

The results indicate that residents in Northern Taiwan have satisfactory family hygiene habits and epidemic prevention consensus. They are strongly cooperative with policies, have high epidemic protection awareness, and wear surgical masks voluntarily. Therefore, they are a major reason for the successful defense against the outbreak. However, because of a discrepancy between the content of epidemic-prevention campaign and policies and their implementation, a flaw has appeared in the line of defense, causing people in Northern Taiwan to exhibit anxiety and headaches. Northern Taiwan exhibits the highest number of confirmed diagnoses in the entire country.

Additional analysis revealed that older adults exhibit the most severe symptoms related to this outbreak. Because of advocacy by news media worldwide, residents aged 51 years or above have paid considerable attention to the epidemic; those aged no higher than 50 years, who frequently use electronic devices because of work or lifestyle, are particularly sensitive to information pertinent to the epidemic. Women in this region are particularly conscious of their personal hygiene and exhibit strong cognitions on diseases as well as self-protection awareness. Therefore, they are particularly strong in epidemic prevention awareness, attitudes, and behaviors and are maintaining their health more easily than men during the epidemic. Those infected with COVID-19 are low in their awareness, attitudes, and behaviors pertinent to epidemic prevention and are prone to insomnia, indigestion, and overeating (Figure 5).

Overall, although flaws exist in the epidemic prevention policy planning and execution in Northern Taiwan, the residents of this region are highly aware of the epidemic and are highly cooperative with epidemic prevention policies. The area is a key to successful epidemic prevention but also a place that can incur an increase in the number of confirmed cases. Correct epidemic prevention measures should be conveyed in the region to maintain the hygiene level of the residents, thereby further enhancing the effectiveness of epidemic prevention.

#### 3.3.2. Central Area

As Table 7 presents, for epidemic prevention attitudes, the respondents from Central Taiwan scored epidemic-prevention campaign and campus consensus the highest (4.2) and personal consensus the lowest (2.7). With awareness, life alertness had the highest score (4.1) and epidemic prevention information gathering the lowest (3.0). Regarding behaviors, voluntarily wearing masks received the highest score (4.0) and personal and community cleaning the lowest (3.6). In physical and mental health, these respondents were the most notable for anxiety (3.2) but least notable for sense of loss (2.0). These results are partially consistent with the findings in [35,36,37].

Further analysis revealed no significant difference between occupations. However, women reported significantly stronger (*p* < 0.01) feelings than men in the following items: for awareness, personal and family habits and consensus, public consensus, health management, and awareness of public epidemic prevention facilities; in attitudes, media information, teachers’ advice, and public measures; and in behaviors, epidemic prevention steps, alertness, epidemic prevention information gathering, and health management. The men reported stronger feelings (*p* < 0.01) for the following items: in physical and mental health, insomnia, headache, indigestion, reduced enthusiasm, and overeating.

The responses significantly differed by age (*p* < 0.01) for the following items: for awareness, personal and family consensus and epidemic prevention acceptance; for behaviors, alertness, teachers’ advice, and public measures; and regarding behaviors, epidemic prevention steps and seeking solutions. Respondents of each age group felt particularly different about the following health-related items: emotions, work ability, enthusiasm, pain sensitivity, insomnia, and sense of loss. Those aged ≤20 years were particularly sensitive to the items related to attitudes, awareness, and behaviors and were particularly prone to be influenced by emotions, work ability, enthusiasm, and pain sensitivity. Those aged 21–50 years reported problems related to work ability, enthusiasm, headache sensitivity, insomnia, and sense of loss.

Differences in physical and mental health were unrelated to awareness, attitudes, or behaviors (*p* > 0.01) but were significantly (*p* < 0.01) related in the following items related to health: emotions and enthusiasm. Moreover, healthy respondents and those diagnosed with chronic disease or having other symptoms differed significantly in the strength of the effects of emotions and enthusiasms.

The results suggest the authorities of the Central Taiwan have actively advocated epidemic prevention policies and campus epidemic prevention. Residents are highly alert against COVID-19 in their daily lives and voluntarily wear masks, contributing to successfully preventing the spread of the epidemic. The flaws exist in that the residents who have weak attitudes regarding personal cleaning, epidemic prevention consensus, and information gathering, and community cleaning has been insufficient. These cause most residents of the region to be anxious and caused 64 citizens in this region to be diagnosed with COVID-19.

As shown in Figure 6. further analysis revealed that residents aged ≤ 20 years, who are highly active and competent at grasping and collecting information from the Internet, are particularly sensitive to items related to epidemic prevention awareness, attitudes, and behaviors. However, because they interact with others frequently, their likelihood of becoming infected with COVID-19 is particularly high. They are more sensitive to emotions, work ability, enthusiasm, and pain than those in other age groups. Women in this region are particularly aware of their personal hygiene and are particularly strong in epidemic prevention awareness, attitudes, and behaviors; therefore, they maintain their health more easily than men during the epidemic. Although most of the residents have remained healthy and actively coordinate with the epidemic prevention measures, their emotions and job enthusiasm are also affected; this is because of industry stagnation and the loss of job opportunities in the face of the epidemic, which impact the residents’ quality of life.

Overall, the residents of Central Taiwan are lacking in their personal cleaning and epidemic prevention consensus, attitudes in gathering information, and awareness of community cleaning. However, the local governments have actively advocated epidemic prevention policies and campus epidemic prevention. The residents are highly alert in their daily lives and wear masks voluntarily. Therefore, this area is a key to successful epidemic prevention but also a spot where an increase in the number of confirmed cases may occur. Improvements must be made in personal cleaning, epidemic prevention consensus, attitudes in gathering information, and awareness of community cleaning to enhance epidemic prevention alertness and behaviors among the residents, thereby reinforcing epidemic prevention.

#### 3.3.3. South Area

As displayed in Table 8, for epidemic prevention attitudes, among the respondents from Southern Taiwan, personal and street visit consensus had the highest score (4.3) and media outreach had the lowest score (3.9). For awareness, the highest score was in life alertness and compliance with public measures (4.2) and the lowest in information gathering (4.0). Regarding behaviors, voluntarily wearing masks was scored highest (4.2) and sense of justice (3.5) lowest. With physical and mental health, these respondents were the most notable for emotion-related problems (3.2) but least notable for suicidal ideation (2.1). These results are partially consistent with the findings in [35,36,37].

Further analysis revealed no significant difference between occupations or physical and mental health status. However, women reported significantly stronger feelings than men for public consensus in the awareness dimension (*p* < 0.01). No significant differences were identified for any other item.

Seeking expert assistance differed significantly by age (*p* < 0.01). Age also influenced scores for the following health-related problems (*p* < 0.01): stress, insomnia, pain sensitivity, and overeating. Those aged ≤20 years were particularly sensitive to the items related to behaviors. Those aged 21–30 years were prone to insomnia, pain sensitivity, and overeating. Those aged 31–40 years experienced stress.

According to the results, the residents of Southern Taiwan have attained high personal and street visit consensus as well as high alertness in daily living. They are willing to comply with public measures and wear masks voluntarily. These are keys to successful epidemic prevention. However, media outreach is insufficient, and the residents lack the attitudes required for gathering information on epidemic prevention and the sense of justice needed to report inappropriate behaviors that inhibit epidemic prevention. These are flaws in the defense against COVID-19; they negatively affect the emotions of most of the respondents in this region and have led to 56 residents being diagnosed with COVID-19.

Further analysis revealed that women in this region are particularly conscious of their personal hygiene and particularly strong in epidemic prevention awareness, attitudes, and behaviors; therefore, they have maintained their health more effectively than men. Those aged ≤20 years, who are highly capable of collecting information from the Internet, are particularly sensitive to epidemic prevention behaviors. However, those aged 21–30 years plan diverse activities and interact with others frequently; consequently, their likelihood of becoming infected with COVID-19 is particularly high. They are also prone to insomnia, pain sensitivity, and overeating. Those aged 31–40 years report higher stress compared with younger respondents, as indicated in Figure 7.

Overall, the media outreach in Southern Taiwan is insufficient, and residents lack the attitudes required to gather information on epidemic prevention and the sense of justice needed to report inappropriate behaviors. However, strong personal and street visit consensus, high alertness in daily living, compliance with public health measures, and voluntary mask wearing have contributed to successful epidemic prevention. Authorities in Southern Taiwan should enhance media outreach, and residents should remind each other to comply with the epidemic prevention measures and actively collect information pertinent to epidemic prevention, thereby improving the epidemic prevention alertness and behaviors of families and individuals and further reinforcing epidemic prevention.

#### 3.3.4. East Area

As Table 9 indicates, for epidemic prevention attitudes, family habits, personal consensus, campus advocacy, and family consensus had the highest score (3.9) and public facilities of epidemic prevention had the lowest score (3.4). With awareness, cooperation awareness and campus alertness ranked highest (4.1) and compliance with public measures (3.2) lowest. Regarding behaviors, the highest score was in compliance with epidemic prevention measures (3.9) and the lowest in community cleaning (3.0). For physical and mental health, anxiety was most notable (3.6) and suicidal ideation least notable (1.8). These results are partially consistent with the findings in [35,36,37].

Further analysis revealed no significant differences between people of different genders, age groups, or occupations. However, a strong correlation was identified between physical and mental health and feelings regarding public measures and media outreach in the awareness dimension (*p* < 0.01). The respondents differed considerably in their pain sensibility according to their reported health status; healthier ones were particularly sensitive to the items related to awareness, whereas those diagnosed as having a cold were particularly sensitive to pain.

According to the analysis, residents in Eastern Taiwan report satisfactory family habits, personal and family consensus, campus advocacy, cooperativeness, and campus alertness. Moreover, they wear masks voluntarily and comply with all epidemic prevention measures. These contribute to the successful epidemic prevention and zero confirmed COVID-19 diagnoses in the region. However, insufficient public epidemic prevention facilities, unsatisfactory attitudes in complying with public measures, and unfavorable community cleanliness are flaws that require improvement.

Further analysis revealed that the local authorities in the Eastern Taiwan emphasize public measures and intensively utilize media to advocate epidemic prevention, contributing to the zero confirmed diagnoses as well as the satisfactory physical and mental health of the residents. However, the region relies on tourism industries and is faced with a reduced number of tourists because of the epidemic, affecting local livelihoods and industries and causing anxiety among residents. Moreover, they are enduring the common cold and high pain sensitivity, as shown in Figure 8.

Again, the residents of Eastern Taiwan exhibit satisfactory family habits, personal and family consensus, campus advocacy, cooperativeness, and campus alertness. Moreover, they wear masks voluntarily and comply with all epidemic prevention measures. These contribute to the successful epidemic prevention in the area. However, public epidemic prevention facilities, unsatisfactory attitudes in complying with public measures, and unfavorable community cleanliness are potential risks in the line of defense against the outbreak. Local authorities must improve epidemic prevention facilities, enforce compliance with the public epidemic prevention measures, and promote community cleaning awareness to further strengthen the epidemic prevention alertness and behaviors of families and individuals and thereby maintain the current status.

#### 3.3.5. Offshore Islands

As presented in Table 10, regarding epidemic prevention attitudes, family epidemic prevention, personal consensus, and public consensus had the highest scores (4.3) and family habits had the lowest score (3.8). For awareness, life and campus alertness received the highest score (4.2) and epidemic prevention knowledge and teachers’ advice the lowest (3.9). With behaviors, the highest score was in compliance with public measures and voluntarily wearing masks (4.1) and the lowest in suggestions from friends and family and community cleaning (3.6). For physical and mental health, anxiety was most notable (3.1) and suicidal ideation the least (1.8). These results are partially consistent with the findings in [35,36,37].

No significant differences were noted between people of different genders, age groups, or occupations. However, a strong correlation was identified between physical and mental health and feelings regarding public epidemic prevention facilities and media outreach in the awareness dimension (*p* < 0.01). The respondents differed considerably in their pain sensibility according to their health status; healthy respondents were particularly sensitive to items related to awareness, whereas those diagnosed as having a cold were particularly sensitive to pain.

Accordingly, satisfactory family epidemic prevention, personal and public consensus, life and campus alertness, and residents’ compliance with public measures and voluntary mask wearing have contributed to the successful epidemic prevention and zero confirmed COVID-19 diagnoses on the offshore islands. However, insufficient family epidemic prevention habits, lack of epidemic prevention knowledge and willingness to seek help from teachers, family, and friends, and unsatisfactory community cleanliness may impede epidemic prevention efforts.

Further analysis revealed that regardless of their gender, age group, or occupation, the residents highly value public epidemic prevention facilities and understand the role of media propagation in depth; these contribute to the successful epidemic prevention and zero confirmed diagnoses as well as to satisfactory physical and mental health. However, the islands, which heavily rely on tourism, are confronted with a reduced number of tourists because of the epidemic, which affects livelihoods and local industries and causes anxiety among the residents. Moreover, the residents are susceptible to headaches, and those diagnosed as having a cold are particularly sensitive to pain (Figure 9).

Once again, satisfactory family epidemic prevention, personal and public consensus, life and campus alertness, and compliance with public measures and voluntary mask wearing have contributed to the successful epidemic prevention on the offshore islands. However, insufficient family epidemic prevention habits along with people’s lack of epidemic prevention knowledge and willingness to seek help from teachers, family, and friends and maintain community cleanliness can impede epidemic prevention efforts. Epidemic prevention and crisis response mechanisms must be enhanced, and epidemic prevention knowledge, family epidemic prevention, and community cleaning awareness must be reinforced to improve epidemic prevention alertness and the behaviors of families and individuals on the offshore islands to reinforce their current epidemic prevention status.

## 4. Conclusions and Recommendations

With regard to epidemic prevention attitudes, awareness, and behaviors, residents of Northern Taiwan have strong family epidemic prevention habits and consensus; those in Central and Southern Taiwan have favorable life and campus alertness related to epidemic prevention; those in Eastern Taiwan and on the offshore islands are highly cooperative with epidemic prevention efforts. Because all citizens are determined to fight the COVID-19 epidemic, their high levels of cooperativeness and voluntarily mask wearing have contributed to the successful epidemic prevention in Taiwan today. However, overall epidemic prevention measures and information delivery plans have been insufficiently comprehensive. The execution of relevant policies has been insufficient in Northern Taiwan; awareness of individual and community cleaning has been insufficient in Central Taiwan. Few in Southern Taiwan have been willing to report inappropriate epidemic prevention behaviors. Public epidemic prevention facilities are insufficient in Eastern Taiwan. Finally, people’s willingness and methods to solve epidemic-related crises have been insufficient and inappropriate on the offshore islands. All these can inhibit epidemic prevention efforts in Taiwan. Residents are psychologically affected by the COVID-19 crisis; most people report emotional instability, anxiety, headaches, indigestion, and overeating habits; some report experiencing psychological problems such as a sense of loss, weakness, and irritability.

Accordingly, we suggest that future research.

(1)Recommendations for Epidemic PreventionUse mass media to convey correct epidemic prevention measures and information. Reinforce personal health habits and encourage people to remind and supervise each other on their habits. Reinforce the execution of administrative decisions in Northern Taiwan. Improve environmental cleanliness in Central Taiwan. Strengthen epidemic prevention consensus in Southern Taiwan. Increase the number of public epidemic prevention facilities in Eastern Taiwan. Last, change the attitudes of offshore island residents in addressing epidemic-related crises.(2)Research SuggestionsFurther explore awareness, attitudes, and behaviors in relation to epidemic prevention among citizens living in various cities and counties along with their physical and mental health based on spatial factors or numbers of confirmed cases;the effects of demographics and behaviors on epidemic prevention awareness, attitudes, and behaviors and on physical and mental health among individuals; and the effects of various exercising behaviors and health habits on people’s resistance to COVID-19; the effects of the COVID-19 pandemic on various industries.

## Figures and Tables

**Figure 1 ijerph-17-04694-f001:**
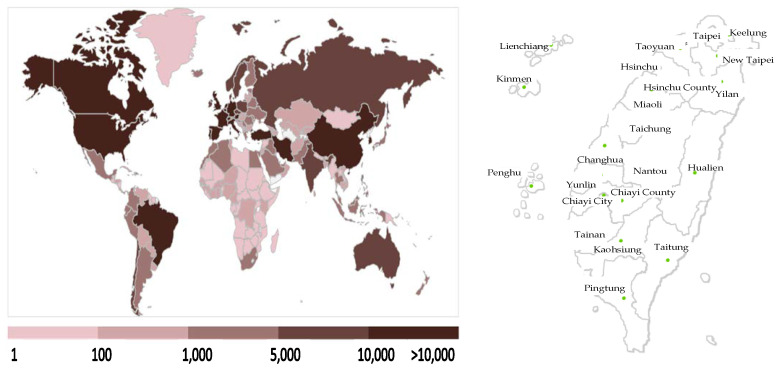
Regions affected by COVID-19 across the world (left) and in Taiwan (up to 9 April 2020).

**Figure 2 ijerph-17-04694-f002:**
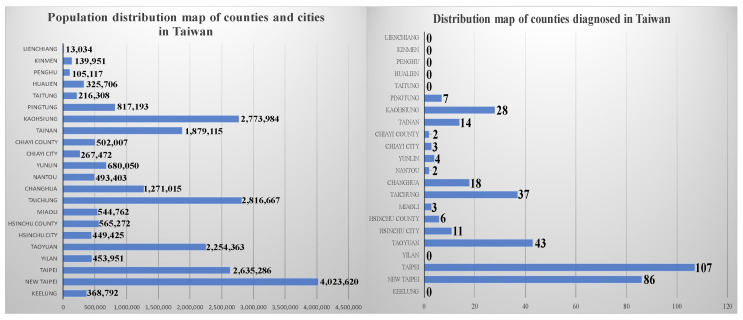
Confirmed cases of COVID-19 across the world and those in Taiwan (up to 9 April 2020).

**Figure 3 ijerph-17-04694-f003:**
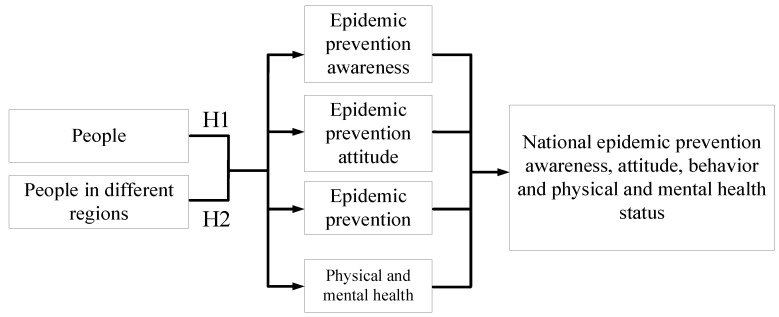
Study framework.

**Figure 4 ijerph-17-04694-f004:**
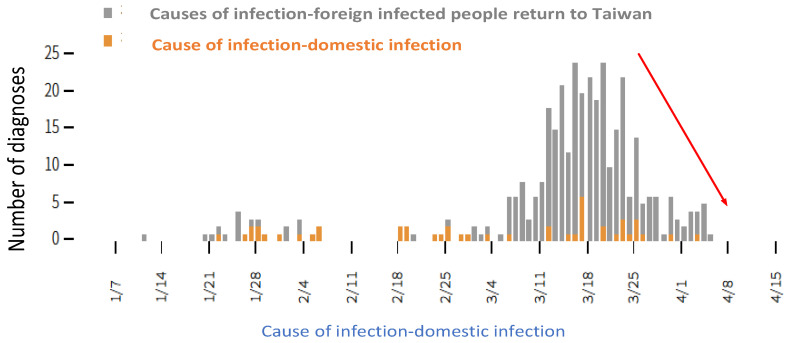
Leading to prevention success in Taiwan.

**Figure 5 ijerph-17-04694-f005:**
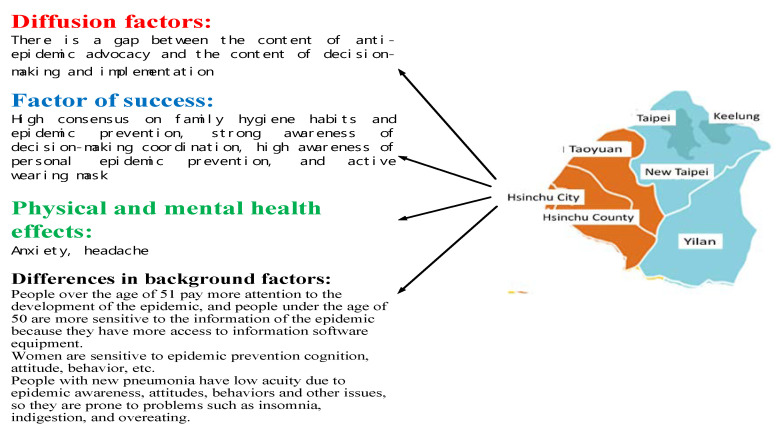
Of key factors for epidemic prevention in north Taiwan.

**Figure 6 ijerph-17-04694-f006:**
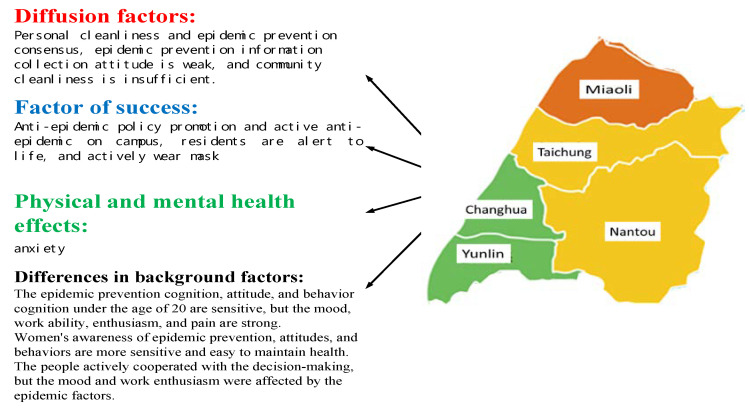
Of key factors for epidemic prevention in central Taiwan.

**Figure 7 ijerph-17-04694-f007:**
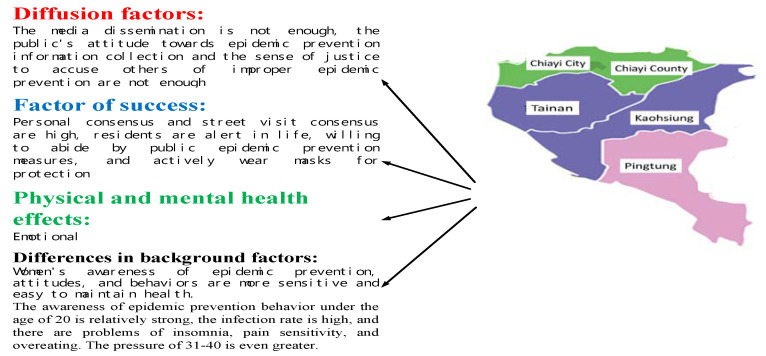
Of key factors for epidemic prevention in southern Taiwan.

**Figure 8 ijerph-17-04694-f008:**
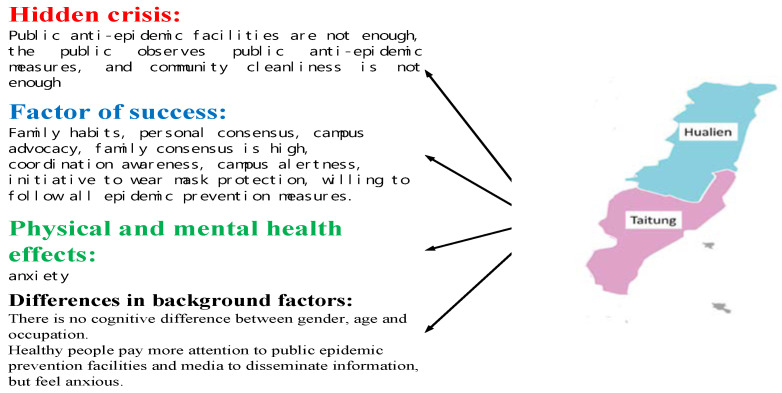
Of key factors for epidemic prevention in east Taiwan.

**Figure 9 ijerph-17-04694-f009:**
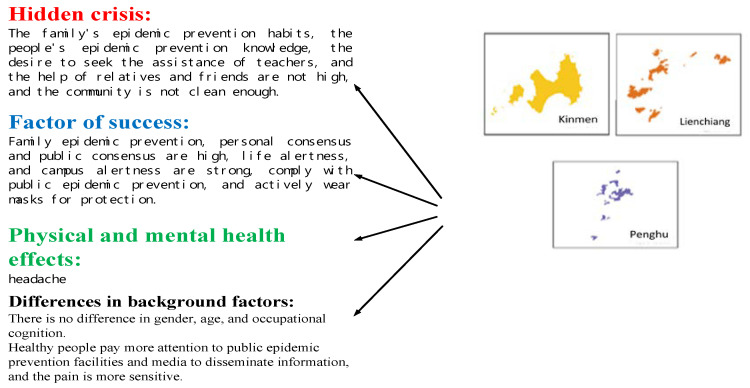
Of key factors for epidemic prevention in offshore islands Taiwan.

**Table 1 ijerph-17-04694-t001:** City and county classification in Taiwan.

Area	County Name	Total
North	Taipei, New Taipei, Keelung, Taoyuan, Hsinchu County, Hsinchu City, Yilan	7
Central	Miaoli, Taichung, Changhua, Nantou, Yunlin	5
South	Chiayi City, Chiayi County, Tainan, Kaohsiung, Pingtung	5
East	Hualien, Taitung	2
islands	Kinmen, Lienchiang, Penghu	3

**Table 2 ijerph-17-04694-t002:** Design on awareness, attitudes, and behaviors in relation to epidemic prevention and the public’s physical and mental health.

Dimension	Subdimension	Subject (M)	Cronbach’α
Physical and mental health	Psychological status (5)	Unstable mood(3.02), feelings of anxiety and panic(3.28), reduced ability(2.67), reduced enthusiasm(2.91), and limited time(2.94)	0.943–0.948
	Spiritual status (5)	Headaches(2.62), fatigue(2.57), pain(2.44), back pain(2.32), and insomnia(2.32)	0.942–0.943
	Attitude and health (6)	Stomach ache(2.25), increased smoking frequency(2.17), irritability(2.36), lack of confidence(2.15), loss of life purpose(2.25), and suicidal thoughts(2.05)	0.942–0.946
Epidemic prevention awareness	Epidemic prevention awareness (4)	Effectiveness of epidemic prevention campaigns(3.88), family influences on individual prevention habits(3.88), formation of prevention habits in families(4.15), and prevention consensus between the public and individuals(4.18)	0.967–0.970
	Consensus on decision-making (4)	Epidemic prevention campaigns(4.18), campus prevention campaigns(4.13), family prevention consensus(4.19), and prevention consensus among interviewees(4.19)	0.966–0.967
	Emergency response (6)	Effect of prevention campaigns on health(3.97), the effects of prevention measures(3.98), the effectiveness of self-protection awareness(3.93), prevention campaigns in public spaces(4.02), prevention campaigns in the media(3.95), and prevention facility installment in public spaces(4.04)	0.967–0.968
Epidemic prevention attitude	Perception (4)	Compliance with prevention measures(4.14), increased prevention vigilance in daily life(4.09), increased campus prevention vigilance(4.13), and the effects of prevention knowledge on infection rate(4.1)	0.959–0.961
	Explicit reaction (4)	Prevention information acquisition(3.99), compliance with teachers’ epidemic prevention advice(4.04), compliance with public space prevention measures(4.07), and informational research of prevention knowledge(3.93)	0.929–0.962
Epidemic prevention behavior	Daily life adaptability (11)	Adequate compliance with the epidemic prevention measures(3.99), active compliance with prevention measures(4.03), correct naming of all prevention processes(3.79), compliance with public space regulations(4.04), wearing of masks(4.08), consultation of experts and scholars for advice(3.92), consultation of online media for advice(3.88), reporting to parents or teachers(3.76), reporting to relatives or friends(3.84), active maintenance of personal hygiene(3.93), active maintenance of cleanliness at home(3.94), medical waste recycling(3.97)	0.967–0.968
	Individual performance (4)	Sharing of prevention knowledge(3.76), reminding persons of inappropriate individual behavior(3.67), active community cleaning(3.67), reminding persons of inappropriate public behavior(3.68)	0.967–0.969

**Table 3 ijerph-17-04694-t003:** Respondent background and interview topics.

Gender	Age	Occupation	Area	Gender	Age	Occupation	Area
male	22	student	northern	male	55	education	central
female	22	Service industry	south	male	38	Tourism	central
male	21	student	northern	female	56	administration staff	northern
male	45	education	islands	male	57	administrative	northern
female	42	financial	south	female	24	student	east
Level	interview topics description						
Cognition	How do you know your epidemic awareness, decision consensus, and resilience? Where does local information generally come from?						
attitude	How do you know your epidemic alert settings and information? Where do local information generally come from?						
behavior	How do you know your personal anti-epidemic attitude and self-protection measures? Where do local information generally come from?						
Physical and mental health	What is the impact of the epidemic on an individual’s physical and mental health? What industry has the greatest impact? Where does the local information generally come from?						
	Based on the above, which one has the greatest impact on people, regions, and occupations? And give your opinion?						

**Table 4 ijerph-17-04694-t004:** Sample analysis.

Area	No.	%	Gender	No.	%	Occupation	No.	%
Northern	711	33.3%	male	842	39.5%	student	1505	70.60%
Central	1129	52.9%	female	1290	60.5%	Restaurant Catering	131	6.10%
South	211	9.9%	**Age**	No.	%	tourism	28	1.30%
East	25	1.2%	Under 20	1068	50.1%	Leisure sport	19	0.90%
islands	58	2.7%	21–30	769	36.1%	Marketing media	13	0.60%
**Health status**	**No.**	%	31–40	105	4.9%	Teachers and civil servants	21	1%
Good health	1960	91.9%	41–50	111	5.2%	Legal administration	13	0.60%
other illnesses	29	1.4%	51–60	57	2.7%	Human Resources	17	0.80%
chronic	53	2.5%	Over 61	22	1%	Finance/Insurance/Accounting	34	1.60%
general	43	2%	**education level**	**No.**	%	Business services	53	2.50%
new coronary	20	0.9%	Elementary	19	0.9%	real estate	23	1.10%
general trauma	27	1.3%	Junior	56	2.6%	Transportation	10	0.50%
**Leisure behavior**	**No.**	%	Senior	301	14.1%	Aviation industry and petroleum industry	2	0.10%
yes	260	12.2%	universities	1697	79.6%	E-commerce	3	0.10%
no	1872	87.8%	Over institute	59	2.8%	Export trade	13	0.60%
**Travel habits**	**No.**	%	**Epidemic prevention message**	**No.**	%	Industrial manufacturing	38	1.80%
yes	1482	69.4%	Oral	118	5.5%	Machining	27	1.30%
no	652	30.6%	Proclamation	42	2%	Agriculture, Forestry and Fisheries	10	0.50%
**Health education knowledge**	**No.**	%	newspapers	105	4.9%	Soldiers and police	22	1%
family	115	5.4%	TV	989	46.4%	Human agency	4	0.20%
School	246	11.5%	Online social platform	878	41.2%	Retail and wholesale	34	1.60%
Government	299	14.0%				Computer communication	22	1%
Mass media	956	44.8%				Performance of art	10	0.50%
Online social platform	516	24.2%				Housekeeping freelance	80	3.80%

**Table 5 ijerph-17-04694-t005:** Analysis of awareness, attitudes, and behaviors in relation to epidemic prevention among Taiwanese nationals and of their physical and mental health statuses.

		Highly Recognized	M	Low Recognition	M	Total Score	Actual Score
Epidemic prevention awareness	Epidemic prevention awareness	prevention consensus between the individuals	4.18	Effectiveness of epidemic prevention campaigns	3.88	20	16.10
	Consensus on decision-making	family prevention consensus	4.19	campus prevention campaigns	4.13	20	16.70
	Emergency response	prevention facility installment in public spaces	4.04	Effect of prevention campaigns on health	3.93	30	23.89
Epidemic prevention attitude	Perception	Compliance with prevention measures	4.14	increased prevention vigilance in daily life	4.09	20	16.45
	Explicit reaction	compliance with teachers’ epidemic prevention advice	4.07	informational research of prevention knowledge	3.93	20	16.03
Epidemic prevention behavior	Daily life adaptability	active compliance with prevention measures and wearing of masks	4.08	reporting to parents or teachers	3.76	55	47.17
	Individual performance	Sharing of prevention knowledge	3.76	active community cleaning	3.67	20	14.77
Physical and mental health	Psychological status	feelings of anxiety and panic	3.28	reduced ability	2.67	25	14.81
	Spiritual status	headaches	2.62	insomnia	2.57	25	12.27
	Attitude and health	irritability	2.36	suicidal thoughts	2.05	30	13.22

**Table 6 ijerph-17-04694-t006:** Of awareness, attitudes, and behaviors in relation to COVID-19 prevention among citizens in Northern area and of their physical–mental health.

Background	Cognition	Attitude	Behavior	Physical and Mental Health
High cognition(M)	Family habits (4.3) Family consensus (4.3) Public measures (4.1)	Cooperation with policies (4.2) Compliance with public measures (4.1)	Voluntarily wearing masks (4.2) Gathering medical waste (4.0) Sharing information (3.8)	Anxiety (3.2) Headache (3.2) Irritability (2.4)
Low cognition(M)	Epidemic-prevention campaign (3.9) Campus epidemic prevention (4.1) Advocacy to promote health (3.9)	Alertness (4.0) Information gathering (3.9)	Conveying epidemic prevention information (3.8) Home cleaning (3.9) Community cleaning (3.6)	Capability (2.6) Lumbago (2.3) Suicidal ideation (2.0)
Different gender (t)	Epidemic prevention advocacy (3.8:4.1) * family habits (3.8:4.1) * self-epidemic prevention (4:4.4) * health protection (3.8:4.0) * public protection (3.8:4.3) *	Media information (3.8:4.1) * teacher suggestions (3.9:4.2) * public measures (3.9:4.3) * search information (3.8:4.1)	Epidemic prevention steps (3.8:4.2) * personal execution (3.8:4.2) * seeking solutions (3.7:4.0) * wear mask (3.8:4.2) * health management (3.8:4.2) *	Mood (3:3.1) * pain sensitivity (2.6:2.4) * eating disorders (2.5:2.2) * lack of confidence (2.4:2.0) *
Different age (F)	Family habits (Under 20,21–30,31–40,41–50 > 51–60) *	Campus epidemic prevention (51–60 > Under 20,21–30) *, teacher suggestions (51–60 > 41–50) *	Teachers (51–60 > 41–50) * relatives and friends for help (51–60 > Under 20,21–30,41–50) * information sharing (51–60 > Under 20,21–30,41–50) * epidemic prevention supervision (51–60 > Under 20,41–50) *	Emotions (21–30,3 > 51–60) * Stress (Under 20,21–30 > 51–60) * pain sensitivity (Under 20,21–30,41–50 > 51–60) * insomnia (41–50 > 51–60) * lack of self-confidence (Under 20,41–50 > 51–60) * seeking death (Under 20 > 21–30) *
Different Health status(F) 1: Good 2: common cold 3: new coronavirus 4. chronic disease 5. common trauma 6. other	public consensus (1,4,2,5,6 > 3) * health protection (1,4,2,5,6 > 3) * public protection and facilities (1,6 > 4,3) *	Campus epidemic prevention (1,2,5,6 > 3) * media information (1,2,6 > 3) * public measures (1,6 > 4,3) *	Epidemic prevention steps (1,6 > 3) * personal execution (1,4,2,5,6 > 3) * seeking solutions (1 > 3) * wear mask and health management (1,4,2,5,6 > 3) * information sharing (1,4,2,5,6 > 3) * epidemic prevention supervision (1,4,2,5,6 > 3) *	pain sensitivity (6 > 1) * insomnia (4,3 > 1,2,5) * indigestion (3 ,6 > 1,5) * eating disorders (3 > 1) * lack of self-confidence (4 > 1) * seeking death (4,6 > 1,2) *

* *p* < 0.01.

**Table 7 ijerph-17-04694-t007:** Of awareness, attitudes, and behaviors in relation to COVID-19 prevention among citizens in Central area and of their physical–mental health.

Background	Cognition	Attitude	Behavior	Physical and Mental Health
High cognition	Epidemic-prevention campaign (4.2) Campus consensus (4.2) Media outreach (4.0)	Life alertness (4.2) Compliance with public epidemic prevention (4.1)	Voluntarily wearing masks (4.0) Consulting the media for advice (3.9) Reminding persons of inappropriate public behavior (3.8)	Anxiety (3.0) Mental weakness (2.6) Irritability (2.7)
Low cognition	Personal consensus (2.7) Advocacy to promote health (3.9) Public measures (3.9)	Information gathering (3.0) Personal cleaning (3.6)	Compliance with public regulations (3.7) Personal cleaning (3.6) Community cleaning (3.6)	Enthusiasm (2.6) Indigestion (2.3) Sense of loss (2.1)
Different gender (t)	personal and family habits (3.5:4.0) * personal (2.9:2.7) * family (3.8:4.3) * consensus with the public (3.8:4.4) * health protection (3.9:4.4) * public protection and facilities (3.7:4.2) *	Media information (3.8:4.3) * teachers ’suggestions (3.7:4.2) * public measures (3.8: 4.2) *	Epidemic prevention steps (3.7:3.9) * personal execution (3.8:4.2) * seeking solutions (3.8:4.2) * health management (3.8:4.2) *	Insomnia (2.4: 2.2) * Pain (2.8: 3.0) * Indigestion (2.8: 2.3) * reduced enthusiasm (2.4:2.3) * eating disorders(2.4:2.1) *
Different age (F)	Personal consensus (Under 20 > 21–30) family consensus (Under 20 > 21–30) * epidemic prevention advocacy acceptance (Under 20 > 21–30) *	Sensitivity (Under 20 > 21–30) teacher suggestions (Under 20 > 21–30) * public measures(Under 20 > 21–30) *	Epidemic prevention steps (Under 20 > 21–30) * seeking solutions(Under 20 > 21–30) *	Emotions (Under 20 > Over 61) * work ability (Under 20,21–30,31–40,41–50 > Over 61) *enthusiasm (Under 20,21–30,31–40,41–50 > 51–60) * pain sensitivity (Under 20 > 21–30,51–60,6) * insomnia (31–40,41–50, Over 61 > 51–60) * lack of self-confidence (41–50 > 51–60) *
Different Health status(F) 1: Good 2: common cold 3: new coronavirus 4. chronic disease 5. common trauma 6. other	NS	NS	NS	Emotions(1,4,6 > 5) enthusiasm(4,6 > 5)

* *p* < 0.01.

**Table 8 ijerph-17-04694-t008:** Of awareness, attitudes, and behaviors in relation to COVID-19 prevention among citizens in Southern area and of their physical–mental health.

Background	Cognition	Attitude	Behavior	Physical and Mental Health
High cognition	Personal consensus (4.3) Street visit consensus (4.3) Public measures (4.1)	Campus alertness (4.2) Compliance with public measures (4.2)	Voluntarily wearing masks (4.2) Community cleaning (3.7)	Emotions (3.2) Mental weakness (2.5) Overeating (2.2)
Low cognition	Epidemic-prevention campaign (4.0) Campus epidemic prevention (4.2) Media outreach (3.9)	Life alertness (4.1) Acquiring information from media (4.0)	Teachers’ reactions (3.7) Sense of justice (3.5)	Ability deterioration (2.6) Lumbago (2.1) Suicidal ideation (1.9)
Different gender (t)	Public consensus (3.7:4.2) *	NS	NS	NS
Different age (F)	NS	NS	Expert advice(Under 20 > 21–30) *	Stress (31–40 > 21–30) * Insomnia (21–30 > Under 20) * pain sensitivity (21–30 > Under 20) * eating disorders(21–30 > Under 20) *

* *p* < 0.01.

**Table 9 ijerph-17-04694-t009:** Of awareness, attitudes, and behaviors in relation to COVID-19 prevention among citizens in Eastern area and of their physical–mental health.

Background	Cognition	Attitude	Behavior	Physical and Mental Health
High cognition	Family habits (3.9) Personal consensus (3.9) Campus advocacy (3.9) Family consensus (3.9) Public consensus (3.7) Advocacy to promote health (3.7) Public epidemic-prevention campaign (3.7) Media epidemic-prevention campaign (3.7) Public epidemic-prevention facilities (3.7)	Cooperation awareness (4.1) Campus alertness (4.1)	Compliance with epidemic prevention measures (3.9) Sense of justice (3.1)	Anxiety (3.6) Headache (2.8) Sense of loss (2.0)
Low cognition	Propaganda and Epidemic Prevention (3.5) Public anti-epidemic facilities (3.4)	Life alertness (3.8) Comply with public anti-epidemic measures (3.2)	Community cleanliness (3.0) Teacher response, reflection from relatives and friends (3.4)	Decreased ability (2.6) Insomnia (1.9) Suicide (1.8)
Different Health status(F) 1: Good 2: common cold 3: new coronavirus 4. chronic disease 5. common trauma 6. other	Public places(1 > 3,2) * media outreach(1 > 3,2) *	NS	NS	pain sensitivity(4 > 1) *

* *p* < 0.01.

**Table 10 ijerph-17-04694-t010:** Of awareness, attitudes, and behaviors in relation to COVID-19 prevention among citizens in Offshore islands and of their physical–mental health.

	Cognition	Attitude	Behavior	Physical and Mental Health
High cognition	Family epidemic prevention, personal consensus, public consensus (4.3) Advocacy to promote health (3.9)	Life alertness, campus alertness (4.2)	Compliance with public measures, voluntarily wearing masks (4.1) Sense of justice (3.9)	Anxiety (3.1) Headache (3.0) Indigestion (2.5)
Low cognition	Family habits (3.8) Campus epidemic prevention (4.2) Public epidemic prevention (4.0)	Epidemic prevention knowledge, teachers’ advice (3.9)	Suggestions from friends and family, community cleaning (3.6)	Sense of urgency (2.9) Insomnia (2.6) Suicidal ideation (2.1)
Different gender (t)	Epidemic Prevention(4.3:3.5) *	NS	Message sharing(3.8:3.9) *	Pain sensitivity (2.6:3.2) * Indigestion (2.2:2.7) * abnormal diet (2.4:2.5) * insomnia (2.4:2.8) * emotional instability (2.3:2.8) * confidence (2.5:2.4) * life goals (2.4:2.5) *
Different age (F)	NS	Media information (Under 20,21–30,31–40,41–50 > Over 61; Under 20 > 31–40) *	Epidemic prevention steps (Under 20,21–30,31–40,41–50 > Over 61) * epidemic prevention supervision (Under 20,21–30,31–40,41–50 > Over 61) *	Pain sensitivity (4 > Under 20,21–30,31–40) * abnormal diet (41–50 > Under 20) *

* *p* < 0.01.

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
