# Peer review of "How to Defend COVID-19 in Taiwan? Talk about People’s Disease Awareness, Attitudes, Behaviors and the Impact of Physical and Mental Health"

_ijerph, 2020, doi:10.3390/ijerph17134694_

Round 1
Reviewer 1 Report
Brief summary
The manuscript entitled “How to defend COVID-19 in Taiwan? talk about people's disease awareness, attitudes, behaviors, and the impact of physical and mental health” evaluates the awareness, attitudes, and behavior in relation to COVID-19 prevention among Taiwanese citizens and their physical and mental health status. To this purpose, an online questionnaire was administered to more than 2,000 people. Specifically, the authors “discussed” their results referring to North, Central, South, East, and Islands areas of Taiwan.
Broad comments
The study is very interesting and novel. Notably, this kind of study could be very useful for decision-makers, national and local governments, who have to find the optimal strategies and measures to face all the issues related to this health emergency. However, the results have to be presented in a more clear way. In addition, there is a lack regarding the discussion of the results.
Specific comments
- Figure 1. The caption shows an image of Taiwan in different colors. In the absence of legend these colors are meaningless. In addition, which is the meaning of the data relating the gender? Are the numbers of infected people or deaths? Therefore, please report this number in the discussion and not in this figure.
- Line 46. “Since the disease outbreak on January 21, 2020, the Taiwanese government has actively adopted various measures to contain the virus spread, instructed its citizens in how to protect themselves, controlled the healthcare supply chain, and encouraged academic institutions to develop antiviral drugs.” Please, add the reference.
- Figure 2. The data reported are incomprehensible. In addition, I suggest reporting the data relating Taiwan population referring to North, Central, South, East, and Islands areas of Taiwan. Finally, which is the meaning of red and black numbers? Which is the meaning of yellow and red underlining?
- Lines 77-78. “To understand an individual’s physical and mental health, researchers should explore his or her physiological, psychological, and spiritual status […]”. Please, specify what you mean with “spiritual status”, the authors should provide a solid theoretical basis.
- Table 1. Please correct “Lslands”.
- Which is the difference between epidemic prevention and epidemic prevention awareness and attitude?
- Section 2.2. Please, report the items of the questionnaire and specify the possible choices of answer of the Likert-scale.
- Line 122. Replace eight with 8.
- Most of the tables are messy and unreadable. Specifically, the authors have to make them again. Therefore, the caption of all tables have to describe the meaning of numbers reported inside.
- Section 3.1. The authors have to report in this section the demographic data of the entire sample related to North, Central, South, East, and Islands areas of Taiwan. The data related to specific occupation and place have to reported in supplementary materials. In addition, the authors have to add also the frequencies for all the data reported. In particular, the number of respondents for each area has to be reported. I suggest reporting in the paper only the data with a percentage higher than the error. In addition, correct the English: education civil servant, petrochemical, performance of art, military fire are meaningless, etc.
- Figure 4. Please, remove the sentences. What is represented on the Y-axis of the figure on the left? Report the labels on both axes. In addition, remove the figure on the right side, since it is redundant with figure 1.
- Chinese writing has to be removed from the whole paper. Specifically, from tables and figures.
- Instead of figures 5-9 please report the results of the questionnaire. Indeed, these figures are meaningless: there is a map with some sentences on the left. In addition, these sentences should be based on results never shown before.
- Please, report tables representing t-test and ANOVA test analysis.
- The discussion of the results should be enriched. The authors should compare their results with other studies on the same topic. As example:
- Motta Zanin, G.; Gentile, E.; Parisi, A.; Spasiano, D. A Preliminary Evaluation of the Public Risk Perception Related to the COVID-19 Health Emergency in Italy. J. Environ. Res. Public Health2020, 17, 3024.
- Taghrir, M. H., Borazjani, R., & Shiraly, R. (2020). COVID-19 and Iranian Medical Students; A Survey on Their Related-Knowledge, Preventive Behaviors and Risk Perception. Archives of Iranian Medicine, 23(4), 249.
- Qiu, J., Shen, B., Zhao, M., Wang, Z., Xie, B., & Xu, Y. (2020). A nationwide survey of psychological distress among Chinese people in the COVID-19 epidemic: implications and policy recommendations. General psychiatry, 33(2).
- Wang, C., Pan, R., Wan, X., Tan, Y., Xu, L., Ho, C. S., & Ho, R. C. (2020). Immediate psychological responses and associated factors during the initial stage of the 2019 coronavirus disease (COVID-19) epidemic among the general population in China. International journal of environmental research and public health, 17(5), 1729.
Author Response
Reviewer 1
Dear reviewer
This article is in response to your suggestions.
Figure 1. The caption shows an image of Taiwan in different colors. In the absence of legend these colors are meaningless. In addition, which is the meaning of the data relating the gender? Are the numbers of infected people or deaths? Therefore, please report this number in the discussion and not in this figure.
Reply Report : Thanks for your suggestions, the error content has been corrected
- 46. “Since the disease outbreak on January 21, 2020, the Taiwanese government has actively adopted various measures to contain the virus spread, instructed its citizens in how to protect themselves, controlled the healthcare supply chain, and encouraged academic institutions to develop antiviral drugs.” Please, add the reference.
Reply Report : Thanks for your suggestions, the error content has been corrected,modified in [8]
Figure 2. The data reported are incomprehensible. In addition, I suggest reporting the data relating Taiwan population referring to North, Central, South, East, and Islands areas of Taiwan. Finally, which is the meaning of red and black numbers? Which is the meaning of yellow and red underlining?
Reply Report : Thanks for your suggestions, the error content has been corrected,modified in Figure 2
Lines 77-78. “To understand an individual’s physical and mental health, researchers should explore his or her physiological, psychological, and spiritual status […]”. Please, specify what you mean with “spiritual status”, the authors should provide a solid theoretical basis.
Reply Report : Thanks for your suggestions, the error content has been corrected,modified in Line . 74
Table 1. Please correct “Lslands”.
Reply Report : Thanks for your suggestions, the error content has been corrected
Which is the difference between epidemic prevention and epidemic prevention awareness and attitude?
Reply Report :
Dear reviewer
This article refers to "anti-epidemic disease" as action, and the epidemic awareness and attitude represent epidemic awareness and awareness.
Section 2.2. Please, report the items of the questionnaire and specify the possible choices of answer of the Likert-scale.
Reply Report : Thanks for your suggestions, the error content has been corrected, modified in Table 2
Line 122. Replace eight with 8.
Reply Report : Thanks for your suggestions, the error content has been corrected
Most of the tables are messy and unreadable. Specifically, the authors have to make them again. Therefore, the caption of all tables have to describe the meaning of numbers reported inside.
Reply Report : Thanks for your suggestions, the error content has been corrected, modified in Table. 7
- Section 3.1. The authors have to report in this section the demographic data of the entire sample related to North, Central, South, East, and Islands areas of Taiwan. The data related to specific occupation and place have to reported in supplementary materials. In addition, the authors have to add also the frequencies for all the data reported. In particular, the number of respondents for each area has to be reported. I suggest reporting in the paper only the data with a percentage higher than the error. In addition, correct the English: education civil servant, petrochemical, performance of art, military fire are meaningless, etc.
Reply Report : Thanks for your suggestions, the error content has been corrected,modified in Table 2 and Page 5
Figure 4. Please, remove the sentences.
Reply Report : Thanks for your suggestions, the error content has been corrected
What is represented on the Y-axis of the figure on the left? Report the labels on both axes. In addition, remove the figure on the right side, since it is redundant with figure 1.
Reply Report : Thanks for your suggestions, the error content has been corrected
Chinese writing has to be removed from the whole paper. Specifically, from tables and figures.
Reply Report : Thanks for your suggestions, the error content has been corrected
Instead of figures 5-9 please report the results of the questionnaire.
Indeed, these figures are meaningless: there is a map with some sentences on the left. In addition, these sentences should be based on results never shown before.
Reply Report : Thanks for your suggestions, the error content has been corrected, modified in Page 4-19.
Please, report tables representing t-test and ANOVA test analysis.
Reply Report : Thanks for your suggestions, the error content has been corrected, modified in Tab 4-15.
The discussion of the results should be enriched. The authors should compare their results with other studies on the same topic.
As example:
- Motta Zanin, G.; Gentile, E.; Parisi, A.; Spasiano, D. A Preliminary Evaluation of the Public Risk Perception Related to the COVID-19 Health Emergency in Italy. J. Environ. Res. Public Health2020, 17, 3024.
- Taghrir, M. H., Borazjani, R., & Shiraly, R. (2020). COVID-19 and Iranian Medical Students; A Survey on Their Related-Knowledge, Preventive Behaviors and Risk Perception. Archives of Iranian Medicine, 23(4), 249.
- Qiu, J., Shen, B., Zhao, M., Wang, Z., Xie, B., & Xu, Y. (2020). A nationwide survey of psychological distress among Chinese people in the COVID-19 epidemic: implications and policy recommendations. General psychiatry, 33(2).
- Wang, C., Pan, R., Wan, X., Tan, Y., Xu, L., Ho, C. S., & Ho, R. C. (2020). Immediate psychological responses and associated factors during the initial stage of the 2019 coronavirus disease (COVID-19) epidemic among the general population in China. International journal of environmental research and public health, 17(5), 1729.
Reply Report : Thanks for your suggestions. Although this article mainly discusses local issues in Taiwan, there are also timely supplements.
Dear Reviewer
Thank you for the above suggestions, I have made corrections based on your suggestions, and hope to have better results.
Many thanks and best regards

Reviewer 2 Report
This manuscript will require extensive re-conceptualisation and revision before it can be considered suitable for publication, in my opinion. It contains, in my opinion, epistemological and methodological flaws. It can be either an exercise in Grounded Theory development or a neutral pursuit of objective truth using Hypothesis Testing and Inferential Statistics. It can not be both.
Methodologically, the use of Cronbach's Alpha Coefficient is not alone a sufficient indicator of scale reliability. It is also not a measure of the validity of the scale.
To be credible, any population survey relating to mental health ought to use a validated instrument. The Kessler Scale of Psychological Distress (K10) is one such scale, and a brief literature search will identify references addressing its validity in the study context.
The presentation of your findings requires more thought. Currently, it is very difficult to both read and interpret most of the Tables. There are occasional appearances of Chinese characters in the text, and there are numerous stylistic and grammatical errors in the manuscript.
A copy of the questionnaire, as a supplementary document to the paper, would also be both useful and appropriate.
There is no doubt that the data you have collected is very important and will be useful to a wider readership. However, the present manuscript, in my opinion, does not enable that outcome.
Author Response
Reviewer 2
Dear reviewer
This article is in response to your suggestions.
Comments and Suggestions for Authors
This manuscript will require extensive re-conceptualisation and revision before it can be considered suitable for publication, in my opinion. It contains, in my opinion, epistemological and methodological flaws. It can be either an exercise in Grounded Theory development or a neutral pursuit of objective truth using Hypothesis Testing and Inferential Statistics. It can not be both.
Reply Report : Thanks for your suggestions.
Methodologically, the use of Cronbach's Alpha Coefficient is not alone a sufficient indicator of scale reliability. It is also not a measure of the validity of the scale.
Reply Report : Thanks for your suggestions, the error content has been corrected
To be credible, any population survey relating to mental health ought to use a validated instrument. The Kessler Scale of Psychological Distress (K10) is one such scale, and a brief literature search will identify references addressing its validity in the study context.
Reply Report : Thanks for your suggestions, the error content has been corrected[32]
The presentation of your findings requires more thought. Currently, it is very difficult to both read and interpret most of the Tables. There are occasional appearances of Chinese characters in the text, and there are numerous stylistic and grammatical errors in the manuscript.
Reply Report : Thanks for your suggestions, the error content has been corrected
A copy of the questionnaire, as a supplementary document to the paper, would also be both useful and appropriate.
Reply Report : Thanks for your suggestions, has been added in due course, in Tab
There is no doubt that the data you have collected is very important and will be useful to a wider readership. However, the present manuscript, in my opinion, does not enable that outcome.
Reply Report : Thanks for your suggestions, I have completed the correction, hope to change your opinion.
Dear Reviewer
Thank you for the above suggestions, I have made corrections based on your suggestions, and hope to have better results.
Many thanks and best regards
Reviewer 3 Report
Dear authors,
The topic is currently pivotal and a population survey in an Eastern country, so close to China and with a very limited number of cases, can be a priori considered very interesting.
I was glad to review it. Several improvements should be done to make clear what are the results and the implications of this research.
Some major comments:
- Methods: the sampling strategy is not clear. Please, describe it, the reference population (size and characteristics), how you randomly selected respondents (did you use a list?) and how you preserved the representativeness of your results (or not).
- Methods: please, describe better how did you collect data. I.e. Did you perform structured or semi-structured interviews? How were the interviewers selected and trained? Where/how were the interviews done (at home, telephone, face to face, ...)?
- Results: did you analyzed possible biases due to the self-selection process and the different kinds of survey administration (interviews vs web)? Did you investigate if the typologies of questions (i.e. on mental health) have produced effects on self-selection?
- Results: too long. What is the key message you want to communicate? I lost the point at the beginning. You should present the results in a more understandable way, considering the different typologies of readers...
- Results: to what is table 7 referring? Please, represent results in a more standard and clear manner.
- Discussions: there are not discussions... It is key to discuss how important is the general public participation in the success of the public health policies, and how to use the public levers to improve the collaborative behaviors of the population (see for instance https://www.cambridge.org/core/journals/behavioural-public-policy/article/behavioural-and-social-sciences-to-enhance-the-efficacy-of-health-promotion-interventions-redesigning-the-role-of-professionals-and-people/01655ECBEE06104DF2D35C61E2A62BC3)
Some minor comments:
- Please, describe the typology of the health system (national vs regional-based, Beveridge or Bismarck...)
- Page 2 Figure 1: please specify the date of data updating and what it is representing (number of cases, number of deaths)
- Page 2 lines 41-54: please specify what kind of measures Taiwan adopted (maybe a brief box with bullet points)
- Page 2 Figure 2: please specify the date of updating of Taiwan data; if it is the same as the other countries, please put the date as the title in the middle and in the figure label
- Pages 7 ss: the table is not well readable
- Pages 20 ss: the table is not well readable, some words are not in the Latin alphabet
Author Response
Reviewer 3
Dear reviewer
This article is in response to your suggestions.
Methods: the sampling strategy is not clear. Please, describe it, the reference population (size and characteristics), how you randomly selected respondents (did you use a list?) and how you preserved the representativeness of your results (or not).
Reply Report : Thanks for your suggestions, the error content has been corrected,modified in Page. 5.
Methods: please, describe better how did you collect data. I.e. Did you perform structured or semi-structured interviews? How were the interviewers selected and trained? Where/how were the interviews done (at home, telephone, face to face, ...)?
Reply Report : Thanks for your suggestions, the error content has been corrected,modified in Table 3 and Page 4-5
Results: did you analyzed possible biases due to the self-selection process and the different kinds of survey administration (interviews vs web)? Did you investigate if the typologies of questions (i.e. on mental health) have produced effects on self-selection?
Reply Report : Thanks for your suggestions, the error content has been corrected,modified in Table 3 and Page 5
Results: too long. What is the key message you want to communicate? I lost the point at the beginning. You should present the results in a more understandable way, considering the different typologies of readers...
Reply Report : Thanks for your suggestions, the error content has been corrected.
Results: to what is table 7 referring? Please, represent results in a more standard and clear manner.
Reply Report : Thanks for your suggestions, the error content has been corrected
Discussions: there are not discussions... It is key to discuss how important is the general public participation in the success of the public health policies, and how to use the public levers to improve the collaborative behaviors of the population (see for instance
https://www.cambridge.org/core/journals/behavioural-public-policy/article/behavioural-and-social-sciences-to-enhance-the-efficacy-of-health-promotion-interventions-redesigning-the-role-of-professionals-and-people/01655ECBEE06104DF2D35C61E2A62BC3)
Reply Report : Thanks for your suggestions, the error content has been corrected,modified in Table 3 and Page 10、14、16、18.
Please, describe the typology of the health system (national vs regional-based, Beveridge or Bismarck...)
Reply Report : Thanks for your suggestions, the error content has been corrected,modified in Page 8
Page 2 Figure 1: please specify the date of data updating and what it is representing (number of cases, number of deaths)
Reply Report : Thanks for your suggestions, the error content has been corrected
Page 2 lines 41-54: please specify what kind of measures Taiwan adopted (maybe a brief box with bullet points)
Reply Report : Thanks for your suggestions, the error content has been corrected
Page 2 Figure 2: please specify the date of updating of Taiwan data; if it is the same as the other countries, please put the date as the title in the middle and in the figure label
Reply Report : Thanks for your suggestions, the error content has been corrected
Pages 7 ss: the table is not well readable
Reply Report : Thanks for your suggestions, the error content has been corrected
Pages 20 ss: the table is not well readable, some words are not in the Latin alphabet
Reply Report :Thanks for your suggestions, the error content has been corrected
Dear Reviewer
Thank you for the above suggestions, I have made corrections based on your suggestions, and hope to have better results.
Many thanks and best regards

Round 2
Reviewer 1 Report
This is an interesting study exploring the awareness, attitudes, and behavior relating to the COVID-19 epidemy in Taiwanese citizens and their physical and mental health status. The results of this research could be relevant to plan prevention measures facing other possible and future epidemy. Therefore, I recommend the publication.
Author Response
Reply to report: Thank you for your suggestion, I wish you all the best.
Dear Reviewer
Dear Reviewer
Thank you for your suggestion.
Many thanks and best regards

Reviewer 2 Report
Thank you for your response to the issues raised in the original review comments. The paper is now more easily readable and more ready for publication. A final effort will be required to address stylistic issues including:
Consistency in the use of capital letters for Table entries and generally in the text, and;
Adequacy of Table column width to ensure the integrity of word structure of headings.
I am also not convinced of the appropriateness of the references to Grounded Theory in relation to the articulation of survey questions (Section 2.2). Grounded Theory is located within a Constructivist onto-epistemology, and its analytical outputs are in the form of theoretical statements. Given that this paper appears to be located within an Empiricist epistemological framework, is it not enough to state that 'survey questions were devised, based on evidence derived from the relevant literature [3-37] These included 14 items...'?
Author Response
Consistency in the use of capital letters for Table entries and generally in the text, and; Adequacy of Table column width to ensure the integrity of word structure of headings.
Reply report: Thank you for your suggestion, the error content has been corrected. For example: P. 5, Table 2.
I am also not convinced of the appropriateness of the references to Grounded Theory in relation to the articulation of survey questions (Section 2.2). Grounded Theory is located within a Constructivist onto-epistemology, and its analytical outputs are in the form of theoretical statements. Given that this paper appears to be located within an Empiricist epistemological framework, is it not enough to state that 'survey questions were devised, based on evidence derived from the relevant literature [3-37] These included 14 items...'?
Reply report: Thank you for your suggestion, the error content has been corrected. For example: P. 4. Lines 111-119.

Reviewer 3 Report
Dear authors,
Thank you for your work on the manuscript. Unfortunately, many concerns on the methodological robustness of your research raised, now that the methods are a little bit more described.
First, the framework is not clear. You wrote that you start from the literature, but what are the theories you refer to?
Second, you assumed that the Cronbach’s α is enough to validate a questionnaire. This is clearly a methodological error. Thus, you should clearly state that the questionnaire is not correctly validated (and put this among the limitations of the study). This is particularly serious in relation to the measurement of reported health status (such as psychological aspects), where there are several validated instruments.
Third, the description of the method is confusing. The sampling, which refers to the questionnaire, is before the on-field interviews that, I guess, were performed before the survey administration in order to construct the framework, as you wrote (p.5 line 128), but the framework is from the literature, is not? On the other hand, you also wrote that "The researchers collected data by conducting field research, individual interviews and questionnaire surveys" (p 3 lines 99-100). Please explain the phases/steps of your study because it is not clear.
Fourth, the paper is too long. I already suggest you cutting the manuscript. You should select what you want to say. What is the message? I missed the point at the third page of results. Please, try to say the essential in no more than 9 pages.
Fifth, I did not find several of the suggestions I provided in the previous round of review. I kindly ask you to carefully check the revisions before re-submitting. I would appreciate it if you provide for the next round of review a detailed report of how you addressed the comments/suggestions (and not only / mainly a report of the fact that you changed the manuscript). This would help in reviewing the paper.
p. 4 line 114: who were the experts? how were they selected? How many people?
p. 5 lines 123-125: I don't see how the impossibility to perform an on-site sampling could have impeded you to randomly select a sample (also using external services such as DoSurvey). This is another crucial point: your sample is done by 70.6% of students. You could rewrite the article focusing only on the 1505 answers of this specific sub-sample, or on the younger people (86% are under 30 years). It would make more sense to do not define this as a population survey, because the respondents are a specific segment of the population.
p. 5 lines 126-127: who were the volunteers? How were they selected? what was the goal of the interviews? on which topics? and what kind of analyses did you performed on 10 observations? it is not clear if the framework is from the literature, or if you integrated it and how, or what else...
p. 7 lines 155-160: this paragraph is a comment that would better fit with the discussions section. Moreover, what kind of analysis supported the statement "The results revealed influential factors leading to the success of prevention measures adopted by the Taiwanese government..."? Any kind of cause-effect analysis was performed and you cannot demonstrate the stated results. The same is repeated in each sub-paragraph of the results (i.e. p. 10 lines 192 and ss).
In general, the Tables are still difficult to read. There is no indication of the meaning of the numbers or the stars. Please, avoid long texts onto a single cell. You could divide the text into different rows, ... or other ways to present your results in a readable way. This is not an effective way to present the results and the paper is not readable and understandable.
I suggest you re-write the manuscript from scratch: this can help you in being more focused.
Author Response
First, the framework is not clear. You wrote that you start from the literature, but what are the theories you refer to?
Reply report: Thank you for your suggestion, the error content has been corrected. For example: P. 4. Lines 111-124.
Second, you assumed that the Cronbach’s α is enough to validate a questionnaire. This is clearly a methodological error. Thus, you should clearly state that the questionnaire is not correctly validated (and put this among the limitations of the study). This is particularly serious in relation to the measurement of reported health status (such as psychological aspects), where there are several validated instruments.
Reply report: Thank you for your suggestion, the error content has been corrected. For example: P. 4. Lines 111-124.
Third, the description of the method is confusing. The sampling, which refers to the questionnaire, is before the on-field interviews that, I guess, were performed before the survey administration in order to construct the framework, as you wrote (p.5 line 128), but the framework is from the literature, is not? On the other hand, you also wrote that "The researchers collected data by conducting field research, individual interviews and questionnaire surveys" (p 3 lines 99-100). Please explain the phases/steps of your study because it is not clear.
Reply report: Thank you for your suggestion, we will put forward more detailed research method description. For example, p.5, line 138-159.
Fourth, the paper is too long. I already suggest you cutting the manuscript. You should select what you want to say. What is the message?
I missed the point at the third page of results. Please, try to say the essential in no more than 9 pages.
Reply to report: Thank you for your suggestion. We will refer to your suggestion and revise the content without affecting the full text description.
Fifth, I did not find several of the suggestions I provided in the previous round of review. I kindly ask you to carefully check the revisions before re-submitting. I would appreciate it if you provide for the next round of review a detailed report of how you addressed the comments/suggestions (and not only / mainly a report of the fact that you changed the manuscript). This would help in reviewing the paper.
Reply report: Thank you for your suggestion, we will put forward the review proposal and reply instructions.
- 4 line 114: who were the experts? how were they selected? How many people?
p。
Reply report: Thank you for your suggestions and content for additional explanation. For example: P. 4. Line 129.
- 5 lines 123-125: I don't see how the impossibility to perform an on-site sampling could have impeded you to randomly select a sample (also using external services such as DoSurvey).
Reply report: Thank you for your suggestion, we will put forward more detailed research method description. For example, p.5, line 148-159.
This is another crucial point: your sample is done by 70.6% of students. You could rewrite the article focusing only on the 1505 answers of this specific sub-sample, or on the younger people (86% are under 30 years). It would make more sense to do not define this as a population survey, because the respondents are a specific segment of the population.
Reply report: Thank you for your suggestion, we will put forward more detailed research method description. For example, p.6, line 163-178.
- 5 lines 126-127: who were the volunteers? How were they selected? what was the goal of the interviews? on which topics? and what kind of analyses did you performed on 10 observations? it is not clear if the framework is from the literature, or if you integrated it and how, or what else...
Reply report: Thank you for your suggestion, we will put forward more detailed research method description. For example, p.5, line 148-159.
- 7 lines 155-160: this paragraph is a comment that would better fit with the discussions section. Moreover, what kind of analysis supported the statement "The results revealed influential factors leading to the success of prevention measures adopted by the Taiwanese government..."? Any kind of cause-effect analysis was performed and you cannot demonstrate the stated results. The same is repeated in each sub-paragraph of the results (i.e. p. 10 lines 192 and ss).
Reply report: Thank you for your suggestion, we will propose an amendment and explain. For example, , page 8, lines 199-200, page 9, lines 222-225, page 9, lines 235-238.
In general, the Tables are still difficult to read. There is no indication of the meaning of the numbers or the stars. Please, avoid long texts onto a single cell. You could divide the text into different rows, ... or other ways to present your results in a readable way. This is not an effective way to present the results and the paper is not readable and understandable.
Reply report: Thank you for your suggestion, we will propose an amendment and explain. For example, Tables 7, 9, 11, 13, 15.
I suggest you re-write the manuscript from scratch: this can help you in being more focused.
Reply report: Thank you for your suggestions, we will propose adjustments and amendments.
Dear Reviewer
Thank you for the above suggestions, I have made corrections based on your suggestions, and hope to have better results.
Many thanks and best regards

Round 3
Reviewer 3 Report
Dear authors,
Unfortunately, I noticed that almost all my suggestions and revisions were not addressed in the current version of the paper.
Please, use my previous revision in case of a further round of review.
Best
Author Response
Dear Reviewer 3
First, the framework is not clear. You wrote that you start from the literature, but what are the theories you refer to?
Reply report: Thank you for your suggestion, the error content has been corrected. For example: P. 4. Lines 121-129.
Second, you assumed that the Cronbach’s α is enough to validate a questionnaire. This is clearly a methodological error. Thus, you should clearly state that the questionnaire is not correctly validated (and put this among the limitations of the study). This is particularly serious in relation to the measurement of reported health status (such as psychological aspects), where there are several validated instruments.
Reply report: Thank you for your suggestion, the error content has been corrected. For example: P. 4. Lines 138-167.
Third, the description of the method is confusing. The sampling, which refers to the questionnaire, is before the on-field interviews that, I guess, were performed before the survey administration in order to construct the framework, as you wrote (p.5 line 128), but the framework is from the literature, is not? On the other hand, you also wrote that "The researchers collected data by conducting field research, individual interviews and questionnaire surveys" (p 3 lines 99-100). Please explain the phases/steps of your study because it is not clear.
Reply report: Thank you for your suggestion, we will put forward more detailed research method description. For example, p.5, line 175-197.
Fourth, the paper is too long. I already suggest you cutting the manuscript. You should select what you want to say. What is the message?
I missed the point at the third page of results. Please, try to say the essential in no more than 9 pages.
Reply to report: Thank you for your suggestion. We will refer to your suggestions and modify the content without affecting the full text description. It has been reduced to 21 pages.
Fifth, I did not find several of the suggestions I provided in the previous round of review. I kindly ask you to carefully check the revisions before re-submitting. I would appreciate it if you provide for the next round of review a detailed report of how you addressed the comments/suggestions (and not only / mainly a report of the fact that you changed the manuscript). This would help in reviewing the paper.
Reply report: Thank you for your suggestion, we have responded according to the content of the last comment and submitted the reply instructions. As explained in the reply letter.
- 4 line 114: who were the experts? how were they selected? How many people?
Reply report: Thank you for your suggestions and content for additional explanation. For example: P. 6. Table 3.
- 5 lines 123-125: I don't see how the impossibility to perform an on-site sampling could have impeded you to randomly select a sample (also using external services such as DoSurvey).
Reply report: Thank you for your suggestion, we will put forward more detailed research method description. For example, p.6, line 175-184.
This is another crucial point: your sample is done by 70.6% of students. You could rewrite the article focusing only on the 1505 answers of this specific sub-sample, or on the younger people (86% are under 30 years). It would make more sense to do not define this as a population survey, because the respondents are a specific segment of the population.
Reply to report: Dear reviewer, thank you for your suggestion.
We have also discovered this phenomenon, but the main purpose of this study is to demonstrate the epidemic awareness and behavior of the people of Taiwan and the impact of their physical and mental health.
Therefore, we will carefully consider your suggestion, and will not affect the main axis of this research topic downwards, which has been revised. If there are deficiencies, it is left to the follow-up research recommendations.
- 5 lines 126-127: who were the volunteers? How were they selected? what was the goal of the interviews? on which topics? and what kind of analyses did you performed on 10 observations? it is not clear if the framework is from the literature, or if you integrated it and how, or what else...
Reply report: Thank you for your suggestion, we will put forward more detailed research method description. For example, p.6, line 185-197.
- 7 lines 155-160: this paragraph is a comment that would better fit with the discussions section. Moreover, what kind of analysis supported the statement "The results revealed influential factors leading to the success of prevention measures adopted by the Taiwanese government..."? Any kind of cause-effect analysis was performed and you cannot demonstrate the stated results. The same is repeated in each sub-paragraph of the results (i.e. p. 10 lines 192 and ss).
Reply report: Thank you for your suggestion, we will propose an amendment and explain. For example, , page 9, lines 253-262, page 9, lines 268-276.
In general, the Tables are still difficult to read. There is no indication of the meaning of the numbers or the stars. Please, avoid long texts onto a single cell. You could divide the text into different rows, ... or other ways to present your results in a readable way. This is not an effective way to present the results and the paper is not readable and understandable.
Reply report: Thank you for your suggestion, we will propose an amendment and explain. For example, Tables 6-10.
I suggest you re-write the manuscript from scratch: this can help you in being more focused.
Reply to report: Thank you for your suggestion.
However, the main purpose and direction of the research investigation are different from this research. Therefore, after adjusting the content to meet your endpoints and not deviating from this research topic, we work hard to make corrections to meet your forward-looking recommendations.
Dear Reviewer
Thank you for the above suggestions, I have made corrections based on your suggestions, and hope to have better results.
Many thanks and best regards
